# Control Technology of Roof-Cutting and Pressure Relief for Roadway Excavation with Strong Mining Small Coal Pillar

Mingzhong Wang [1], Hanghang Zheng [2,*], Zhenqian Ma [2] , Hang Mu [2] and Xiaolei Feng [2]

1   Guizhou Panjiang Refined Coal Co., Ltd., Liupanshui 553533, China
2   School of Mining, Guizhou University, Guiyang 550025, China
*   Correspondence: zhh1878696@163.com

**Abstract:** In order to solve the problem of serious deformation and failure of surrounding rock and difficult maintenance of gob-side entries with strong mining-induced small coal pillars, taking the A110607 return airway of the Shanwenjiaba Coal Mine as the engineering background, the key parameters of roof-cutting and pressure relief control technology for roadway excavation with strong mining-induced small coal pillars were studied by using two-way concentrated blasting roof-cutting and pressure relief technology, combined with theoretical analysis, numerical simulation and a field industrial test. A collaborative control scheme of "roof-cutting pressure relief + anchor cable combined support" is proposed. The test results show that when the height of roof-cutting is 8 m, the angle of roof-cutting is 15°, and when the width of the coal pillar is 3 m, the effect of roof-cutting and pressure relief is the best. Through the field blasting test, it was determined that the blast hole spacing was 800 mm, 321 charge structure was used in the intact roof, and 221 or 211 charge structures were used in the broken roof and geological structure zones. During the driving and strong mining period, the roof and floor movement of the roof-cutting section of the roadway excavation was reduced by about 38% compared with the uncut section, and the deformation of the two sides of the roadway was reduced by about 44% compared with the uncut section. It shows that the collaborative control scheme of "roof-cutting pressure relief + anchor cable combined support" has a good effect on the roadway excavation driving of small coal pillars in strong mining.

**Keywords:** strong mining; numerical simulation; roadway excavation; roof-cutting and pressure releasing; blasthole spacing

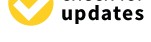



## 1. Introduction

In recent years, roadway excavation with small coal pillars has been widely used in China because of its high recovery rate, long service life, simple distribution of stress characteristics and effective isolation of mined-out areas [1–3]. With the increase of mining depth, the occurrence conditions of coal seam are complex. The buried depth of gob-side entry is large, the ground stress is increased, and the influence of mining is strong, which leads to severe mine pressure in the process of excavation and working face mining. In severe cases, there are too many cracks in small coal pillars, which cannot play a bearing role and collapse [4–6].

At present, although roadway excavation has solved the problems of coal mine recovery rates, air leakage in mined-out areas, coal gangue spontaneous combustion, high gas pressure and coal spontaneous combustion, and gas coupling disasters [7–9], the problem of severe mine pressure in roadway excavation and working face mining has not been effectively solved. The key influencing factor of mine pressure behavior in roadway excavation is the fracture position of the mined-out area roof. The appropriate fracture position has an important influence on the stability of the surrounding rock structure [10–12]. Roof-cutting pressure relief can artificially determine the main roof-breaking position, to achieve active

control of mine pressure. Therefore, how to effectively control the deformation of surrounding rock and determine the reasonable width of coal pillar is the biggest problem faced by small coal pillar roadway excavation. Aiming at the deformation and failure mechanism and control technology of surrounding rock in roadway excavation, a large number of scholars achieved rich research results [13–15]. Taking the transportation roadway of 11,808 working faces in the Qinglong Coal Mine as the engineering background, Wang et al. [16] studied the key parameters of roadway excavation by using roof-cutting and pressure relief two-way shaped charge blasting technology, combined with theoretical analysis, numerical simulation, and field measurements, and achieved good results in an industrial test. In view of the problems of the low recovery rate of large coal pillars and severe mine pressure in retaining small coal pillars in the Fusheng Coal Mine of Lu'anGroup, Zhang et al. [17] used the theoretical analysis, numerical simulation and field measurement of mine pressure to realize the stability of surrounding rock in the heading stage and mining stage of this working face. In order to solve the problem of roadway deformation in the gob-side entry of the Zhangshuanglou coal mine, Zhang [18] studied the technology of blasting roof-cutting and roadway protection by means of theoretical analysis and field tests, and put forward the corresponding countermeasures. Bie et al. [19] put forward the technology of advanced pre-splitting roof-cutting and pressure relief roadway excavation for the difficult control of surrounding rock of high-stress roadway excavation in the deep mine of the ZhaoguNo.1 Coal Mine, and carried out a field industrial test to effectively control the deformation of surrounding rock of the roadway.

Although many scholars have performeda lot of research on the theory and control measures of surrounding rock in roadway excavation, they have achieved fruitful results [20–22]. However, in recent years, with the improvement of mining intensity and the tension of mining replacement, the phenomenon of face-to-face mining has often occurred. Under the influence of strong mining, there are few studieson the technology of strict control of small coal pillars with roof-cutting and pressure relief. Therefore, this paper takes the A110607 return airway of the coal mine of Guizhou Panjiang Coal and Electricity Group as the research background, and carried out research on roof-cutting and pressure relief control technology of strong mining small coal pillar roadway excavation.

## 2. Engineering Background

### 2.1. Roadway Overview

The thickness of the coal seam in the A110607 working face is 1.2–3.1 m, with an average thickness of 2.5 m and an average dip angle of 8°, which is a preparation working face. The lower part is Coal Seam 7#, and the existing working face A110705 is about 9 m away from the 6# layer. The A110607 return airway is driven along the mined-out area and along the roof of the 6# coal seam. The total work quantity is 921.5 m, and the buried depth is about 190 m. The location of the working face is shown in Figure 1. The roof and floor strata of the coal seam are mainly composed of mudstone and argillaceous siltstone. The roof and floor strata of the coal seam are shown in Figure 2. In order to reduce the amount of excavation and support costs, while recycling coal resources as much as possible, the return airway (trapezoidal roadway width × medium height = 5.2 m × 2.8 m) was used to implement small coal pillar roof-cutting pressure relief.

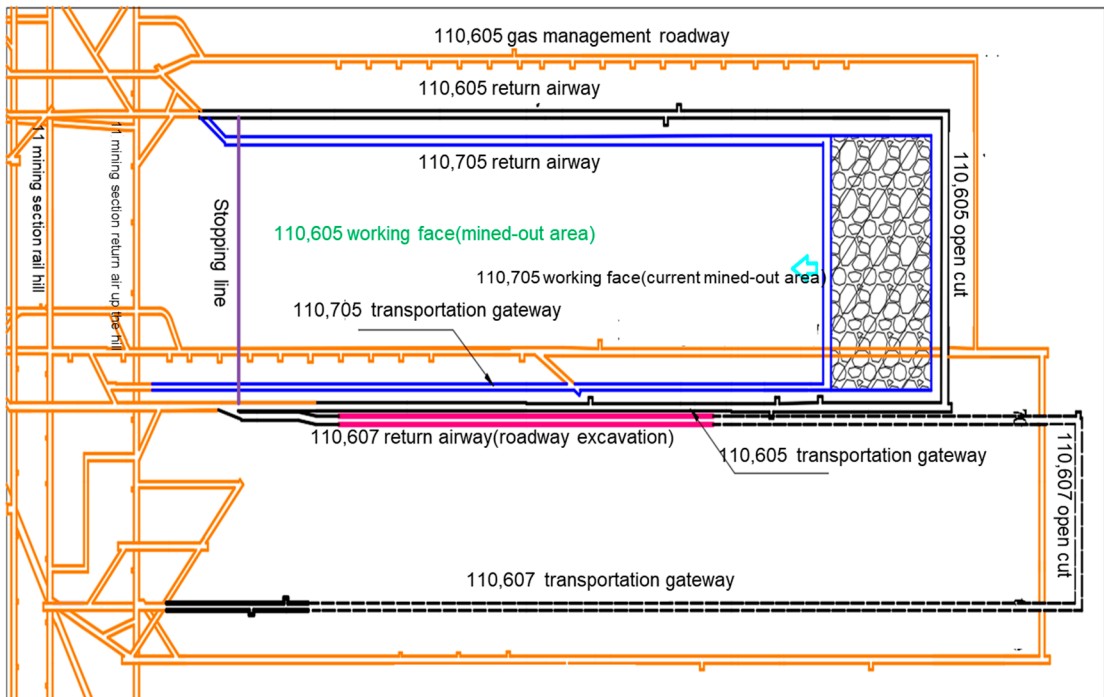

**Figure 1.** Working face layout diagram.

| seam | thickness/m | histogram | Lithologic description |
|---|---|---|---|
| | 10.08−11.4 | | The upper part is gray, and the middle layer is siltstone. Contains argillaceous and charcoalous strips, see a small amount of biological debris; The lower part is dark gray, and the local black-gray thick bedded flint-bearing limestone. The flint is a lumpy limestone with argillaceous siltstone, dividing the limestone, and the lithology of the upper and lower layers is thin and thick, and the stratum is stable throughout the region. |
| | 2.01−5.11 | | Grey, thin to medium layered. Convergence oblique layer, turbid layer. There are several layers of argillaceous rhombus rock in the middle. The upper part is sandwiched with a layer of unstable gray limestone about 0.40m thick, which gradually transitions downward to argillaceous rock, containing biodebris, which is unstable. |
| | 1 | | Grey mudstone with a small band of mirror coal. |
| 6# | 2.52 | | Black, powdery, chipped. According to the actual disclosure of the two alleys of the working face and the cutting eye, the structure of the coal seam is complex, the upper layer is fragmented, 0~2.1m thick, the lower layer is powdered, 0.5~2m thick, and the lower layer contains 0.1~0.3m sandwiches. The total thickness of the coal seam is 0.8~4.2m, and the general thickness is 2.52m. |
| | 0.19−0.99 | | |
| | 4.8−5.62 | | Mudstone, light gray, gray, clumpy, containing fossilized plant roots. |
| | | | Dark gray siltstone - light gray fine sandstone, medium to thick layered. Parallel intermittent wave-like stratigraphic development. The downward spiral gradually overflows into grey argillaceous siltstone. |
| | 0.95−1.34 | | Light gray limestone, medium to thick layered, fine crystalline, argillaceous content aggravated downward. |
| 7# | 1.23−1.46 | | Grey-black, lumpy, columnar. It is mainly a single coal seam. |
| | 1.89−4.92 | | Sandy mudstone, grey, topped by a 0.5 m thick light grey lump containing fossilized plant roots, with a thin layer of coal or coal lines at the bottom. |

**Figure 2.** Comprehensive column.

*2.2. Analysis on Failure Characteristics of Gob-Side Roadway*

Combined with the mining conditions and geological conditions of the mine, the main failure factors of the roadway are analyzed as follows:

(1) Strength of surrounding rock: the roadway is driven along the 6# coal seam. The surrounding rock is mainly mudstone, with low strength. It is very easy to break and expand under the action of multiple stresses, which is the main factor leading to the deformation and damage of the roadway.

(2) Mining influence of working face: A110607 return airway is strongly affected by tunneling disturbance and A110705 working face, which leads to severe deformation and failure of roadways and is the key factor of roadway deformation and failure.

(3) The roadway support is unreasonable: the original support method of the roadway adopts the traditional bolt-mesh-spurting support, the prestress of the full thread bolt is low, and the initial support cannot provide high support resistance. At the same time, under the influence of dynamic pressure, the working load of the bolt cable decreases after the loosening and failure of surrounding rock in the shallow part of roadway, which leads to the insufficient overall bearing capacity of the roadway and cannot effectively control the deformation of roadway.

## 3. Theoretical Calculation of Small Coal Pillar Width
*Modelling*

The smaller the size of the small coal pillar within a reasonable range, the less the deformation of the surrounding rock of the roadway, and it is easier to effectively control the coal pillar and roadway to improve the coal recovery rate. However, if the coal pillar is too narrow and the strength of the coal pillar is too low, it is easy to cause the coal pillar to be destroyed in a large area and lose its stability, which leads to the bolt running through the whole coal pillar into the goaf and the support strength beingreduced. Therefore, the reasonable minimum coal pillar width B can be calculated by the following formula:

$$B = X_1 + X_2 + X_3 \tag{1}$$

In the formula, $X_1$ is the plastic zone damage range generated in the coal body on the side of the goaf, and its value is calculated according to the following formula [20]:

$$X_1 = \frac{mA}{2tan\varphi_0} ln \left[ \frac{K\gamma H + \frac{C_0}{tan\varphi_0}}{\frac{C_0}{tan\varphi_0} + \frac{P_Z}{A}} \right] \tag{2}$$

In the formula, $m$ is the thickness of coal seam; $A$ is the lateral pressure coefficient; internal friction angle of $\varphi_0$ coal; $C_0$ is the cohesion of coal; $K$ is the stress concentration factor; $\gamma$ is the average bulk density of overlying strata; $H$ is the depth of roadway; $P_z$ is the support resistance of the coal side; $X_2$ is the effective length of the side bolt; and $X_3$ is the abundance of coal pillar width.

According to the geological data of the A110607 return airway in theWenjiaba Coal Mine, the main theoretical parameters of the working face are as follows: the average thickness of coal seam m = 2.5 m, the lateral pressure coefficient A = μ/(1 − μ) = 0.35, μ is 0.26, the internal friction angle of coal $\varphi_0$ = 24°, the cohesion of coal $C_0$ = 1.8 MPa, the stress concentration coefficient K = 1.6, the average bulk density of overlying strata $\gamma$ = 25 kN/m$^3$, the buried depth of roadway H = 190 m, and the support resistance $P_z$ of support to coal side is 0 on the side of goaf.

The plastic zone damage range $X_1$ = 1.03 m generated in the coal pillar of the upper section goaf is calculated. The effective length of the bolt is taken as $X_2$ = 1.5 m, and the reserved abundance $X_3$ of the coal pillar width is generally calculated as 0.2 of ($X_2 + X_3$), so $X_3$ = 3.09 m, and the reasonable coal pillar width is taken as B = 3 m.

## 4. Determination of Key Parameters of Roof Cutting and Pressure Relief

### 4.1. Theoretical Analysis of Cutting Height

The roof-cutting height not only affects the filling height of the goaf and the supporting effect on the overlying rock mass, but also affects the caving form of the roof rock mass in the goaf. Reasonable roof-cutting height can cut the roof rock mass in a certain range of goaf and fill the goaf, which plays a good supporting role in the overlying rotary hinged rock mass. At the same time, the height of roof-cutting should be high enough to cut off the key strata and achieve the purpose of roof-cutting and pressure relief. In order to make the roof collapse in the range of roof-cutting in goaf play an effective supporting role in overlying strata, the height of cutting seam can be calculated according to the theory of rock mass expansion according to the following formula:

$$H = \frac{H_m - \Delta H_1 - \Delta H_2}{K - 1} \tag{3}$$

$H_m$ is the mining height of coal seam, m; $\Delta H_1$ is the roof subsidence, m; $\Delta H_2$ is the amount of floor heave, m; $K$ is the bulking coefficient of rock mass.

According to the lithology thickness of each layer of the roof, the weighted calculation is carried out coefficient K is 1.4, without considering the roof subsidence and floor heave, the maximum mining height H is 3.1 m, and H = 7.75 m is calculated.

Comprehensive comparison of A110605 transport roadway roof lithology histogram, the direct roof of the roadway is gray mudstone, containing a small amount of specular coal strip, thickness 2.01–5.11 m, the main roof is the third lower limestone of the standard, the general thickness is 10.8–11.4 m, the lithology is hard, the thickness is 4.3–10.6 m, and the average thickness is 7.8 m. In order to make the roof of the goaf in the range of roof-cutting easier to collapse along the end of the pre-splitting cutting seam, the height of the initial roof-cutting is determined to be 8.0 m, and the later adjustment is made according to the peeping results of the borehole.

### 4.2. Theoretical Analysis of Cutting Angle

According to the reference [23], because the roof-cutting angle is not obvious to the roof-cutting effect of the inclined coal seam, the angle change is no longer set to study the influence of the roof-cutting angle, so the fixed roof-cutting angle is selected as 15°.

## 5. Numerical Simulation

### 5.1. Modelling

In order to study the influence of mined-out area roof caving and roadway surrounding rock stability under different roof-cutting heights and different coal pillar widths of roadway excavation under strong mining, according to the engineering geological background of the A110607 return airway, the numerical calculation model is established by using discrete element software UDEC. The model size is 100 m × 60 m, as shown in Figure 3. The bottom boundary of the model and the surrounding are constrained, and the top is a free boundary, applying 4.8 MPa vertical stress (uniform load). The Mohr–Coulomb model is used for rock blocks and the Coulomb slip model is used for joints. The physical and mechanical parameters of simulated coal strata are shown in Table 1. The contact characteristic parameters of blocks are shown in Table 2.

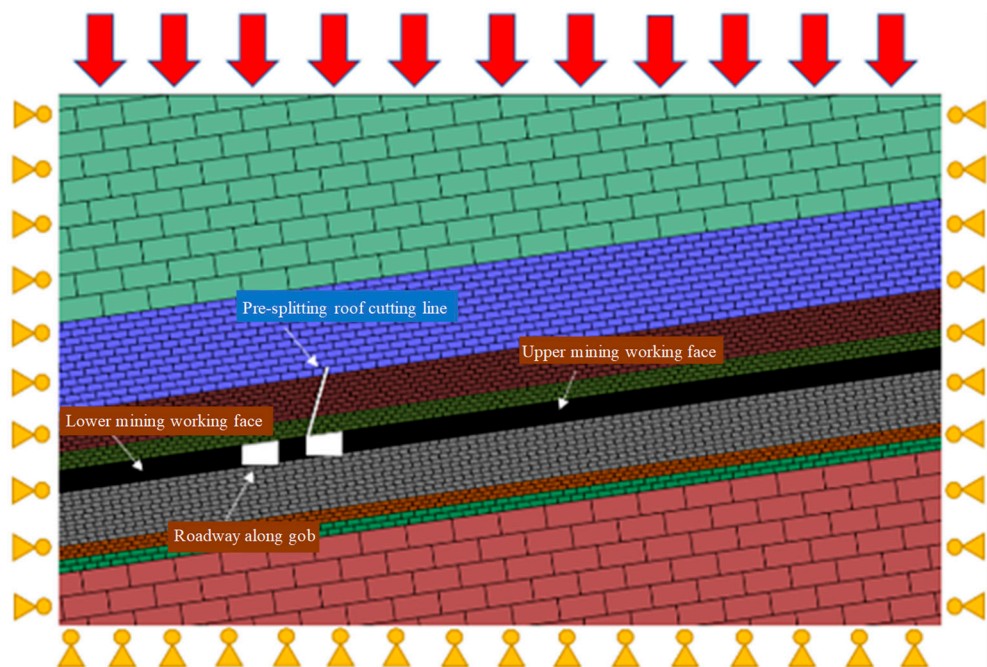

**Figure 3.** Numerical model.

**Table 1.** Each layer of rock physical and mechanical parameters in model.

| Rock Formation | Density/ (g·cm$^{-3}$) | Bulk Modulus/ (GPa) | Shear Modulus/ (GPa) | Force of Cohesion/ (MPa) | Angle of Internal Friction/(°) | Tensile Strength/ (MPa) |
|---|---|---|---|---|---|---|
| Mudstone | 2550 | 3 | 1.3 | 1.7 | 29 | 1.5 |
| Grey mudstone | 2500 | 1.5 | 0.7 | 1.2 | 23 | 0.3 |
| 6#coal seam | 1400 | 1.9 | 0.9 | 0.8 | 27 | 0.3 |
| Siltstone | 2600 | 5.8 | 4.2 | 3.2 | 33 | 1.8 |
| Limestone | 2650 | 22 | 14 | 3 | 39 | 2.9 |
| 7#coal seam | 1440 | 1.9 | 1.3 | 0.8 | 30 | 0.3 |
| Muddy siltstone | 2550 | 4.1 | 3.1 | 3.2 | 35 | 1.8 |

**Table 2.** Mechanical properties of model contact surface.

| Rock Formation | Density/ (g·cm$^{-3}$) | Kn/(GPa) | Ks/(GPa) | Force of Cohesion/(MPa) | Angle of Internal Friction/(°) | Tensile Strength/(MPa) |
|---|---|---|---|---|---|---|
| Mudstone | 2550 | 2.7 | 12 | 0.8 | 28 | 0 |
| Grey mudstone | 2500 | 1.9 | 8 | 0.8 | 22 | 0 |
| 6#coal seam | 1400 | 1.2 | 5.8 | 0.3 | 22 | 0 |
| Siltstone | 2600 | 2.7 | 1.2 | 0.8 | 28 | 0 |
| Limestone | 2650 | 6 | 19 | 1 | 38 | 0 |
| 7#coal seam | 1440 | 1.9 | 8 | 0.6 | 28 | 0 |
| Muddy siltstone | 2550 | 2.6 | 13 | 0.9 | 29 | 0 |

*5.2. Numerical Calculation Scheme*

Combined with the actual situation of the site, the simulated excavation process is: pre-splitting cutting line, return airway and transport roadway of upper working face → excavation of coal seam of upper working face → excavation of gob-side entry and data detection. In order to better analyze the influence of mining, this paper first analyzes the influence of roadway surrounding rock under different roof-cutting height, and then analyzes the influence of different coal pillar width. Since the effect of roof-cutting angle

on inclined coal seam is not obvious [24], the influence of roof-cutting angle is no longer studied, so the roof-cutting angle of 15° is selected as the fixed angle. According to the theoretical analysis, the roof-cutting height wasselected as 4 m, 6 m, 8 m and 10 m, and the coal pillar width wasselected as 2 m, 3 m, 4 m and 6 m. The simulation scheme is shown in Table 3. By excavating slits of different heights and setting different widths of coal pillars, the effects of different heights of roof-cutting and widths of coal pillars on the stability of gob-side entry surrounding rock are analyzed.

**Table 3.** Numerical simulation scheme design.

| Scheme | A | B | C | D | E | F | G |
|---|---|---|---|---|---|---|---|
| Roof cut height/m | 4 | 6 | 8 | 10 | 8 | 8 | 8 |
| Coal pillar width/m | 3 | 3 | 3 | 3 | 2 | 4 | 6 |
| Support pattern | The supporting method remains unchanged | | | | | | |

*5.3. Stress and Displacement Analysis of Surrounding Rock of Gob-Side Roadway under Different Cutting Height*

In order to study the influence of different roof-cutting heights on the stability of surrounding rock, the simulation and data processing of schemes A, B, C and D are carried out respectively, as shown in Figures 3 and 4.

(1) Figure 4 is the displacement cloud and roadway surrounding rock deformation histogram under different roof-cutting height. As can be seen from Figure 4, when the roof-cutting height is 4 m, due to the small height of the roof-cutting, the overburden strata in the mined-out area are separated, and the rock blocks rotate to form a hanging roof, which does not completely cut off the stress transfer of the overburdened strata, resulting in the deformation of the roadway surrounding rock and the coal pillar gradually increasing, which has an adverse effect on the maintenance of the roadway surrounding rock. At this time, the maximum deformation of roadway roof is 413 mm, and the deformation of solid coal wall and coal pillar wall are 130 mm and 316 mm, respectively. When the roof-cutting height is 6 m, the roof-cutting range is within the range of the immediate roof. After the roof-cutting, the roof of the mined-out area collapses and sinks along the pre-splitting cutting seam line. However, at this time, the immediate roof does not completely cut down. Instead, a large separation layer is generated between the immediate roof and the overlying strata, which is in a hanging roof state. Once it collapses, it will affect the stability of the roadway. At this time, the maximum deformation of roadway roof is 333 mm, and the deformation of solid coal wall and coal pillar wall is 70 mm and 238 mm respectively. When the roof-cutting height is 8 m, the cutting line runs through the direct roof to part of the basic roof. After the roof-cutting, the roof of the mined-out area collapses smoothly along the cutting line, and the caving rock can effectively support the basic roof of the rotation. The maximum deformation of the roof is 307 mm, the movement of the solid coal wall is 55 mm, and the movement of the coal pillar is 206 mm; when the roof-cutting height increases to 10 m, the caving range of the overlying strata in the mined-out area increases, and the basic roof can fall smoothly along the slit line. Compared with the roof-cutting height of 8 m, the maximum deformation of the roof, coal pillar wall and solid coal wall does not change much, and the overall deformation of the roadway is small, which is beneficial to the maintenance of the roadway. It is found that increasing the roof-cutting height has little effect on the stability of the roadway.

(2) Figure 5 shows the stress cloud, the stress curve of solid coal wall and the maximum stress diagram of coal pillar under different roof-cutting heights. It can be seen from Figure 5 that when the roof-cutting height is 4 m, the peak stress of the solid coal wall is 15.8 MPa, and the peak stress is about 2.2 m from the solid coal wall of the roadway, and then gradually decreases and tends to be stable. The hanging roof formed by the uncollapsed immediate roof and the main roof on the side of the coal pillar leads to a large range of high stress areas in the coal pillar, resulting in stress concentration. The peak stress of the coal

pillar is 8.7 MPa. The large stress concentration will make the coal pillar unable to support the weight of the overlying strata, thus a large area of piece, coupled with the impact of strong mining, the roadway is prone to large amounts of deformation and instability, which brings difficulties to the maintenance of the roadway. When the roof-cutting height is 6 m, the peak stress of the solid coal wall is 15 MPa, which is consistent with the 4 m, and the mechanical transmission of the mined-out area roof is not completely cut off, resulting in the increase of the separation of the immediate roof and the main roof above the mined-out area. When the roof-cutting height reaches 8 m, the stress connection above the coal wall and above the coal pillar is weakened, and the peak stress is reduced. The peak stress is 14.2 MPa and 7 MPa, respectively. The peak stress concentration in the solid coal wall is 2.2 m–2.5 m away from the roadway side. When the height of roof-cutting increases to 10 m, it has little effect on the stress distribution because of the basic collapse.

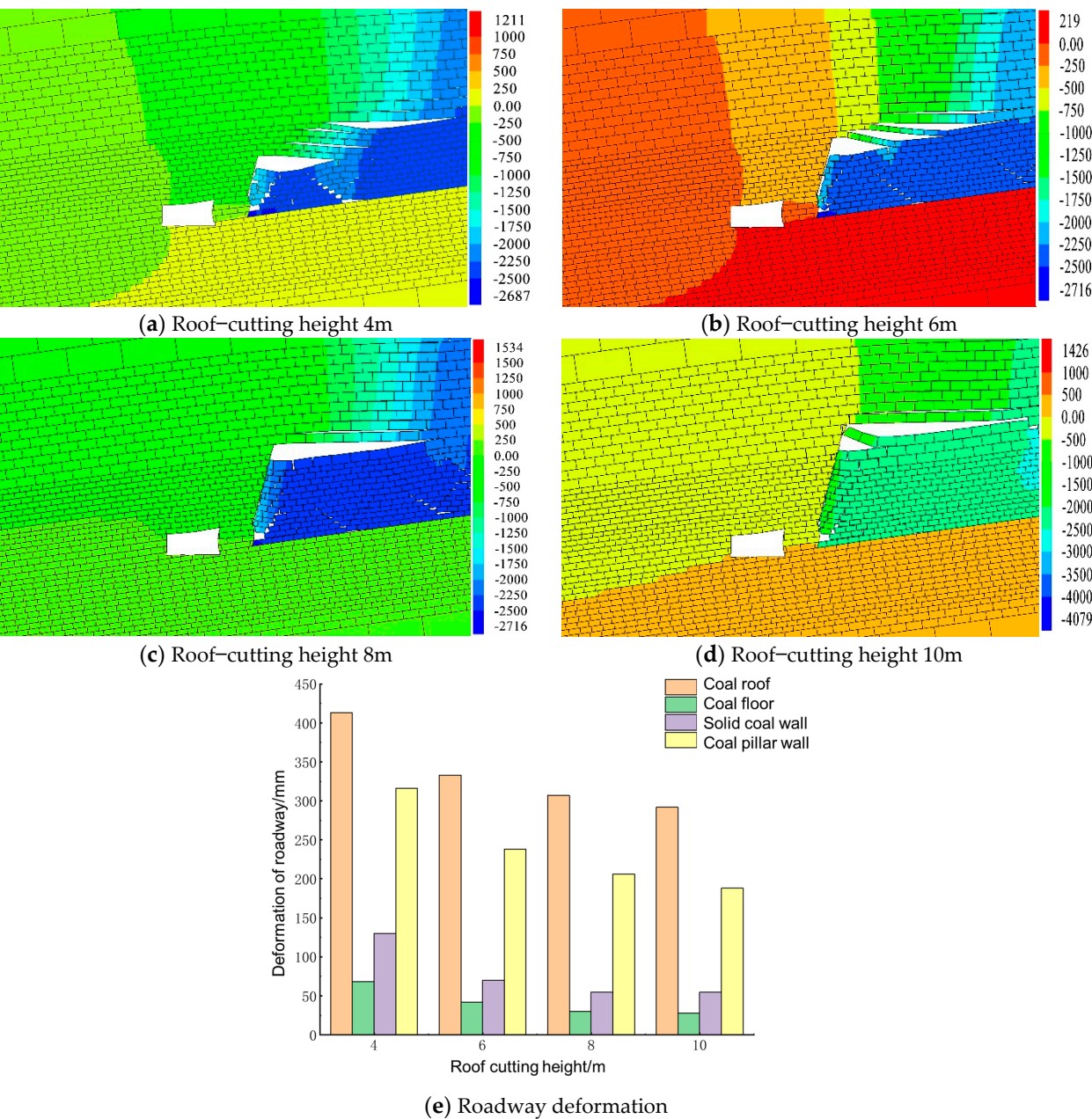

**Figure 4.** Displacement cloud and deformation histogram of roadway surrounding rock under different roof-cutting heights.

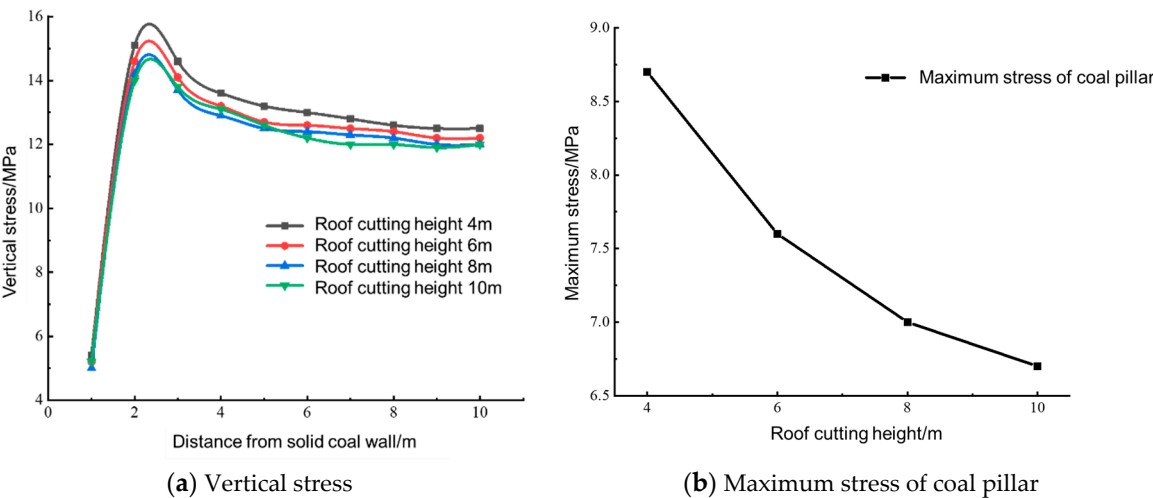

(**a**) Vertical stress　　　　　　　　　　　(**b**) Maximum stress of coal pillar

**Figure 5.** Stress cloud diagram and solid coal wall stress curve and maximum stress diagram of coal pillar under different roof-cutting height.

In summary, from the simulation results, it can be seen that when the roof-cutting height is 4 m, 6 m and 8 m, the roof has different degrees of hang, separation and collapse. When the roof-cutting height is 8 m, the direct roof completely collapses better than 6 m. From the data analysis, the increase of the roof-cutting height is easy to reduce the peak stress of the solid coal wall, the peak area moves to the deep, and the peak stress of the coal pillar also decreases, which is conducive to the stability and maintenance of the roadway. When the height of roof-cutting is increased to 10 m, the influence of the height of roof-cutting on the roadway is not obvious. Therefore, according to the simulation, the optimal height of roof-cutting is 8 m.

*5.4. Stress and Displacement Analysis of Surrounding Rock of Gob-Side Roadway under Different Coal Pillar Width*

From the above analysis, it can be seen that when the roof-cutting height is 8 m, the roof-cutting effect is the best. Therefore, in order to study the influence of different coal pillar widths on the stability of roadway surrounding rock, the schemes C, E, F and G are simulated and data processed respectively, as shown in the figure.

(1) Figure 6 shows the displacement cloud and the deformation histogram of roadway surrounding rock under different coal pillar widths. It can be seen from the figure that along the direction of roadway roof and floor: with the increase of coal pillar width, the roof deformation decreases first, then increases and then decreases, showing a wave-shaped change. When the width of coal pillar is 2–3 m, the deformation of roof decreases from 352 mm to 307 mm, and when the width of coal pillar is 3–4 m, the deformation decreases by 30 mm. The deformation of the floor also decreases first and then increases. When the width of the coal pillar is 2–3 m, the deformation decreases from 62 mm to 30 mm, and when 3–6 m, it shows an increasing trend, which is not conducive to roadway maintenance. Along the direction of the roadway side: the deformation of the solid coal wall decreases first and then increases with the increase of the width of the coal pillar, showing a step wave change. When the width of the coal pillar is 2–3 m, the deformation is reduced from 85 mm to 55 mm, and the deformation is not obvious when the width of the coal pillar is 3–6 m; the deformation of the coal pillar wall gradually decreases and changes linearly. When the width of the coal pillar is 2–3 m, the deformation increases from 213 mm to 206 mm. When the width of the coal pillar is 3–6 m, the deformation gradually decreases to 76 mm.

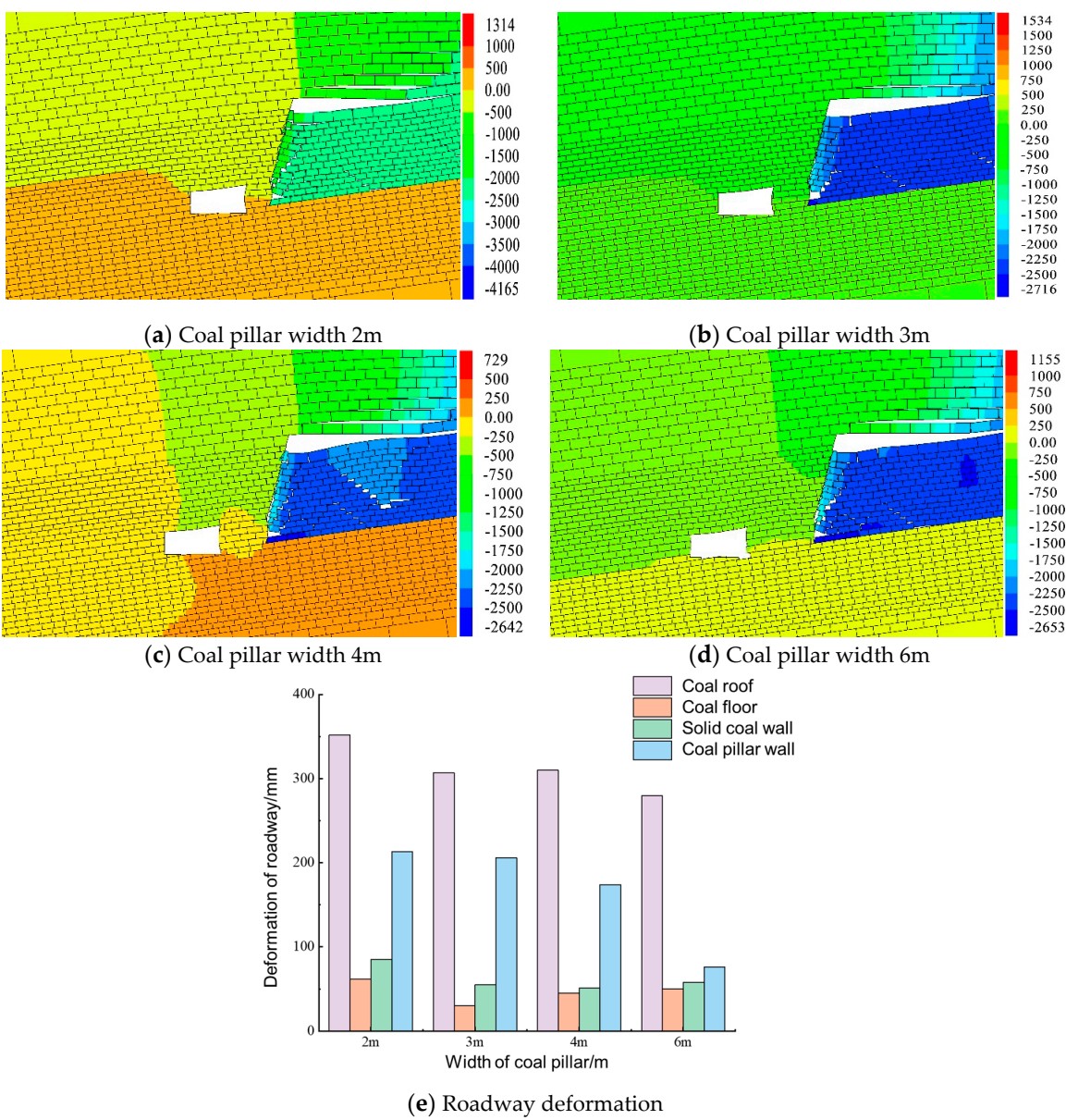

**Figure 6.** Displacement cloud and roadway surrounding rock deformation histogram under different coal pillar widths.

(2) Figure 7 is the stress cloud diagram under different coal pillar widths, the stress curve of the solid coal wall and the maximum stress diagram of the coal pillar. It can be seen from the figure that the vertical stress distribution of coal wall in gob-side entry is obviously different when the width of coal pillar changes. The stress curve of the solid coal wall along the boundary direction of the model shows a trend of increasing first and then decreasing. The difference is that the distance from the peak stress of the solid coal wall to the roadway side is different. When the width of the coal pillar is 2 m, the peak stress position is about 2 m away from the solid coal wall of the roadway, and the peak value is 19 MPa, while the vertical stress peak (5 MPa) of the coal pillar is much lower than that of the solid coal wall, indicating that the roof strata are mainly carried by the solid coal wall. When the width of coal pillar is 3 m, the peak stress of coal wall is 14.2 MPa, 2.3 m away from the solid coal wall of roadway, and the peak stress of coal pillar is 7 MPa. When the coal pillar width is 4 m, the peak stress position is about 2 m away from the solid coal wall of the roadway, the peak value is 15.7 MPa, and the peak stress of the coal pillar wall (11 MPa); when the width of the coal pillar increases to 6 m, the vertical stress in the coal

pillar increases rapidly and exceeds that of the solid coal wall, and the vertical stress peaks are 14.5 MPa and 16.3 MPa, respectively. At this time, the overburden load is transferred from the solid coal wall to the coal pillar wall, resulting in the vertical stress peak of the coal pillar wall beinghigher than that of the solid coal wall, and the roadway beingin a high-stress environment.

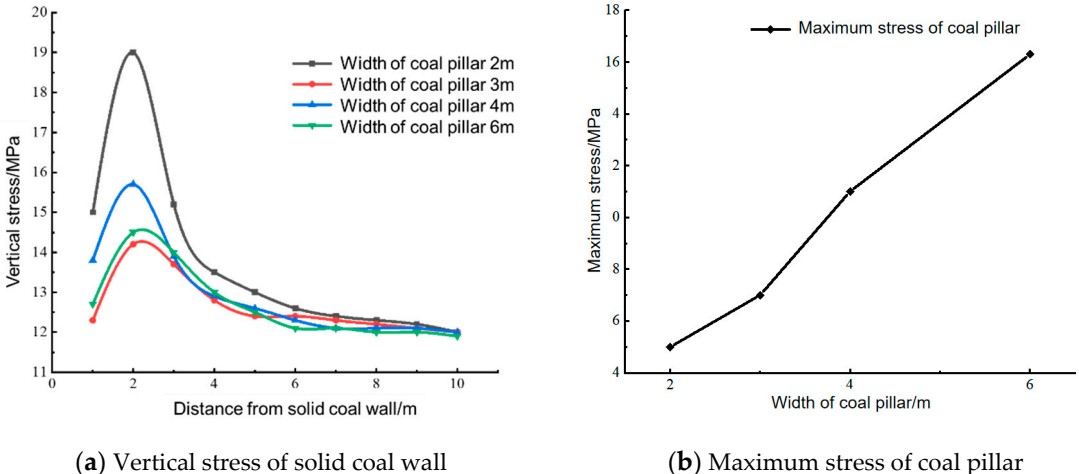

(**a**) Vertical stress of solid coal wall   (**b**) Maximum stress of coal pillar

**Figure 7.** Stress cloud and solid coal wall stress curve and maximum stress diagram of coal pillar under different coal pillar widths.

In summary, when the width of the coal pillar is 3 m, it is beneficial to the maintenance of the roadway. If the width of the coal pillar is too small, the deformation and piece of the coal pillar will be serious. Although the width of coal pillar is too large to alleviate the stress distribution on coal pillar and solid coal wall, it is easy to cause a lot of waste of coal resources. Finally, the reasonable width of coal pillar is 3 m.

## 6. Field Engineering Applications

According to the previous numerical simulation, considering the site construction conditions, construction economic factors, resource conservation, stability and safety of roadway excavation and so on, the roof-cutting height is determined to be 8 m and the coal pillar width is 3 m.After the key parameters of roof-cutting are determined, the bilateral cumulative explosion is used to cut the roof. In order to effectively guarantee the stability of surrounding rock in roadway excavation, the cooperative control scheme of "roof-cutting and pressure relief + bolt-cable combined support" is adopted to improve the stress state of surrounding rock to the greatest extent and improve the stability of surrounding rock.

### 6.1. Pre-Splitting Roof Cutting Blasting Parameters

To explore the pressure relief effect after pre-splitting roof-cutting blasting. Determination of reasonable blast hole spacing, explosive quantity and charge structure and other blasting parameters, the scene through the 500 mm, 600 mm, 700 mm, 800 mm, 900 mm five different blast hole spacing with 321 charge structure using coal mine allowable three-stage emulsion explosive (specification 35 mm × 300 mm/roll, each roll weight 333 g) with energy collecting tube, tube length 1500 mm, outer diameter 42 mm, inner diameter 36.5 mm, pre-splitting blasting test, the use of charge for pre-splitting blast hole. The cutting effect was observed by borehole peeping in the holes without charge between blasting holes. The charge structure is shown in Figure 8.

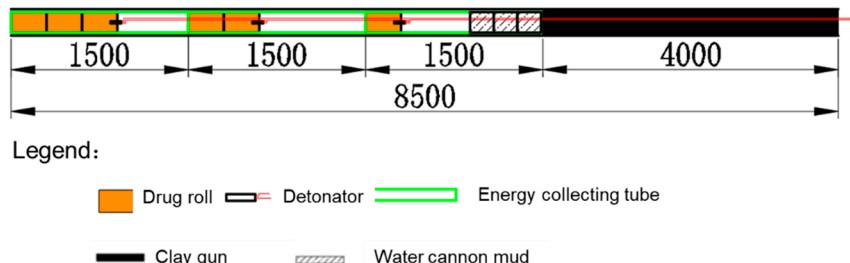

**Figure 8.** The321 charge structure diagram.

After the field test, according to the test results, the integrity of the roof after blasting is good, and the orifice is not collapsed and deformed by blasting impact. When the blasthole spacing is 500 mm, 600 mm, 700 mm and 800 mm, there are obvious conductive cracks in the peephole. The cracks in the hole are symmetrically distributed and relatively straight, and the crack rate is above 85%. When the blasthole spacing is 900 mm, a conductive crack appears in a peephole, and the crack rate is only 15%.Finally, the blasthole spacing is determined to be 800 mm, the 321 charge structure is used in the complete roof position, and the 221 or 211 charge structure is used in the broken roof and geological structure zone. Borehole peeping is shown in Figure 9.

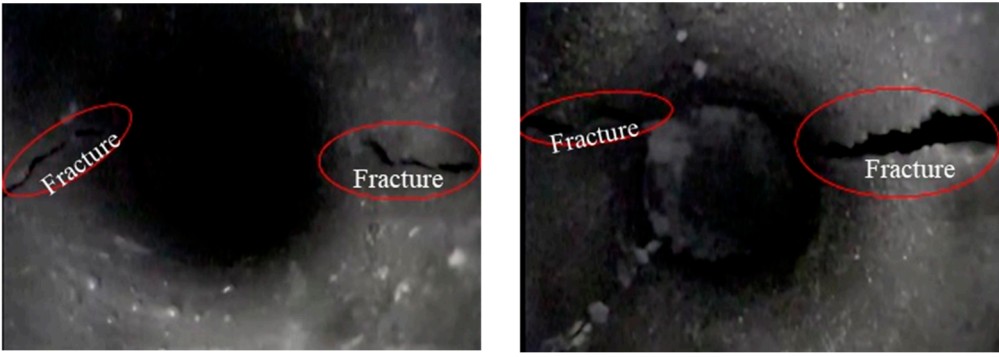

**Figure 9.** Fracture diagram in hole with 800 mm spacing.

*6.2. Roadway Support Parameters*

As shown in Figure 10, the roof is supported bythe "bolt + steel mesh + anchor cable + W steel strip", and the bolt is supported by left-handed non-longitudinal reinforcement thread bolt ($\varphi$ = 20 mm, L = 2500 mm). The row spacing between bolts is 800 × 800 mm; the anchor cable adopts $\varphi$ = 21.6 mm, L = 8300 mm, and the row spacing between anchor cables is 1600 × 800 (1600) mm; each piece of the W steel strip is 4.5 m long and supported with anchor cable. The anchor cable is constructed in the steel strip hole. The $\Phi$ = 6.5 mm steel mesh is used, the specification is 1700 × 900 mm, the grid is 100 × 100 mm, the lap length of the steel mesh is ≥100 mm, and the interval between the lap parts of the steel mesh is 200 mm. The two strands of No.14 wire are tied and fixed.High side using "I steel + steel mesh" support, I steel length 3.1 m, spacing (center distance) is 0.8 m, steel mesh $\Phi$ = 6.5 mm, specification 1700 × 900 mm, grid is 100 × 100 mm. The low side is supported by anchor net. The anchor rod is glass fiber reinforced plastic composite anchor rod ($\varphi$ = 20 mm, L = 2000 mm), and the row spacing is 800 × 800 mm. Each anchor rod adopts two MSCK2335 resin anchoring agent as end-anchoring. Reinforcement mesh with $\Phi$ = 6.5 mm, specification is 1700 × 900 mm, the grid is 100 × 100 mm, reinforcement mesh lap length is 100 mm, reinforcement mesh lap position interval 200 mm using two strands of 14 wire tied fixed.

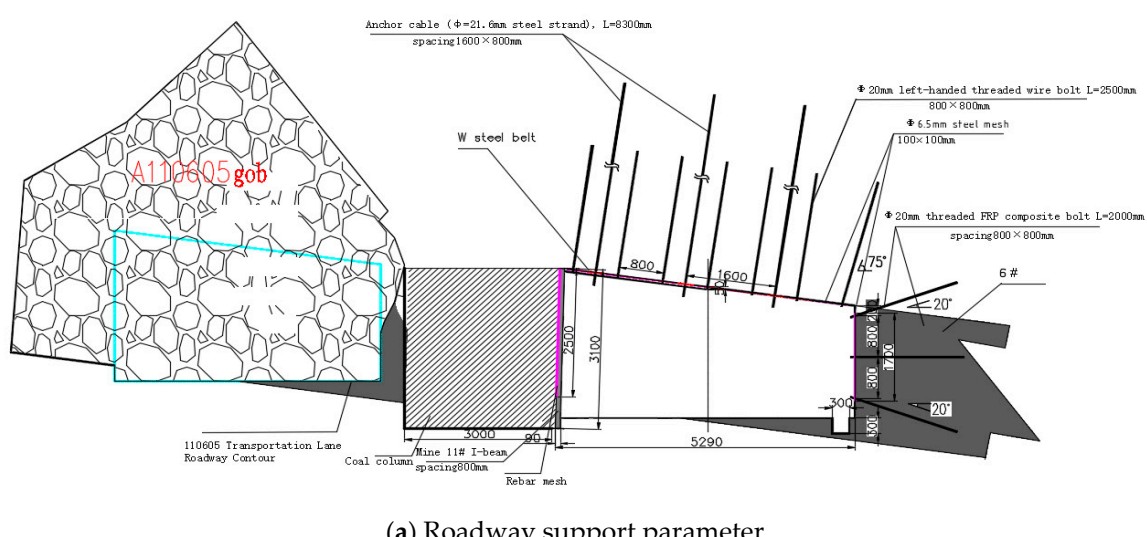

(**a**) Roadway support parameter

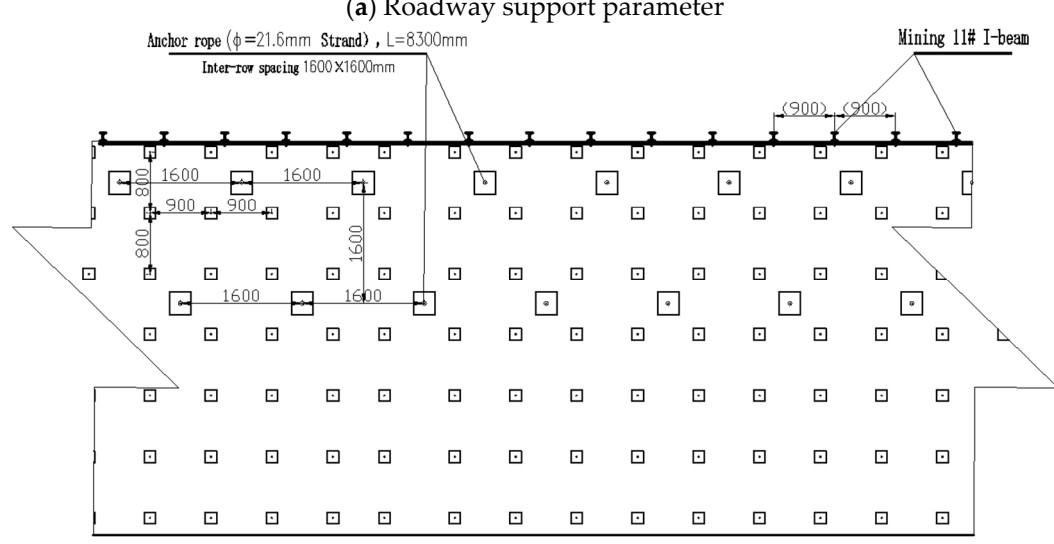

(**b**) Support plane diagram

**Figure 10.** Roadway support parameter.

### 6.3. Roadway Control Effect

In the roof-cutting and uncut sections of the excavated roadway in the A110607 return airway, it is divided into two parts for monitoring. Starting from the position of the entry test roadway section, one station is added for each 50 m excavation, two stations in the roof-cutting section, and two stations in the uncut section. The cross-point placement method is used to monitor the distribution of roadway points. The monitoring results are shown in Figure 11. It can be seen from the figure that the deformation of roadway is related to the deformation time during the roadway excavation with roof-cutting and pressure relief. The deformation of two sides and roof of A110607 return airway increases nonlinearly. In 0–70 days, the deformation of two sides of 3# station in the uncut section of roadway increases from 0 to about 450 mm, and the deformation of roof and floor increases from 0 to about 300 mm. The deformation of the two sides of the 4# station increases to about 400 mm, and the deformation of the roof and floor increases from 0 to about 350 mm. After that, the deformation of the two sides and the roof and floor of the A110607 return airway gradually stabilized, and the deformation trend of the roof and floor and the two sides of the roadway gradually slowed down. In the range of the roof-cutting section, because the roadway is cut off by the roof of the mined-out area side to form a stable structure, the stress of the roof of the A110607 return airway is reduced, and the deformation of the roadway is weakened. Under the action of roof-cutting and pressure relief, the deformation of the roadway is

greatly reduced, and the maximum deformation of the two sides is maintained at about 240–280 mm, which is about 44% of the deformation of the roadway without roof-cutting. The deformation of roof and floor increases from 0 mm to about 200 mm, which is about 38% of the deformation of roadway without roof-cutting. In general, after the improvement of roof-cutting pressure relief and support mode, the surrounding rock control of A110607 return airway in the roof-cutting pressure relief section during roadway excavation is better. Compared with the non-roof-cutting section, the deformation of surrounding rock is small, and the effect of roof-cutting pressure relief along the mined-out area is better. The roadway control effect is shown in Figure 12.

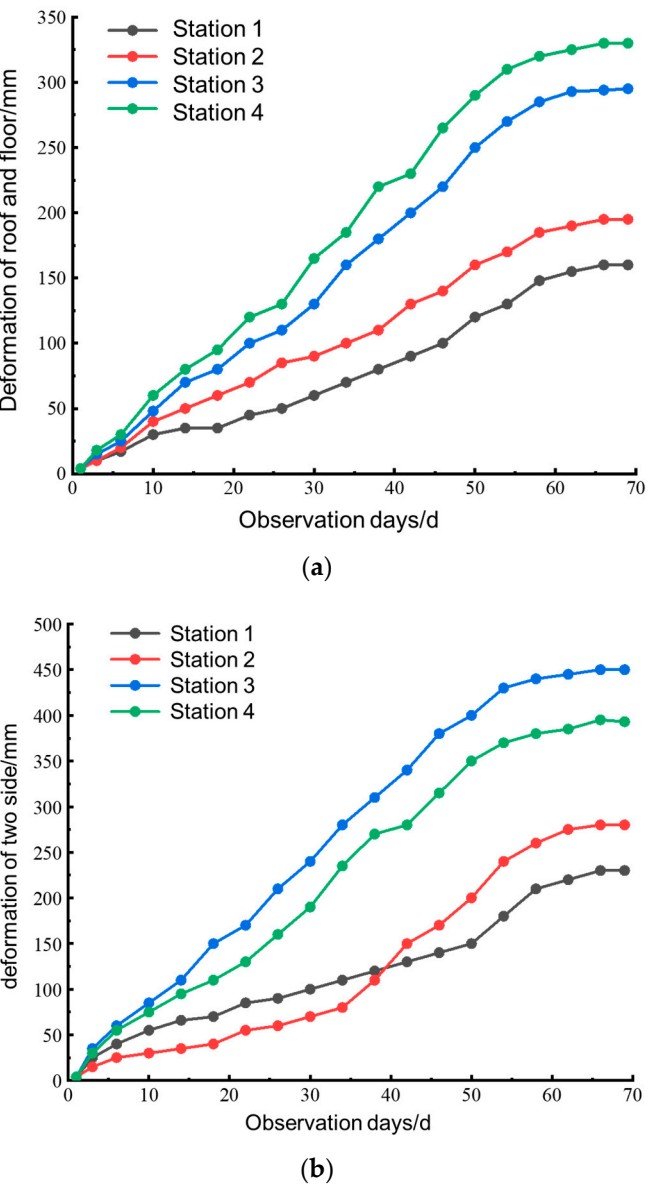

(**a**)

(**b**)

**Figure 11.** Roadway surface displacement monitoring diagram. (**a**) The approximate amount of the roof and floor of the roadway of the measuring station; (**b**) the two sides of the roadway of the measuring station moved close.

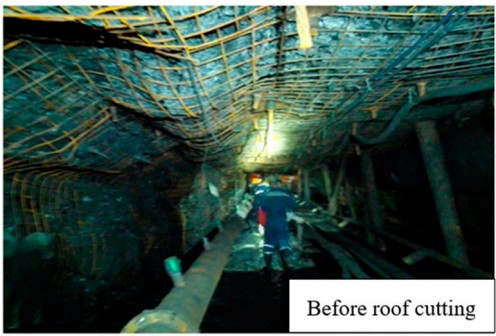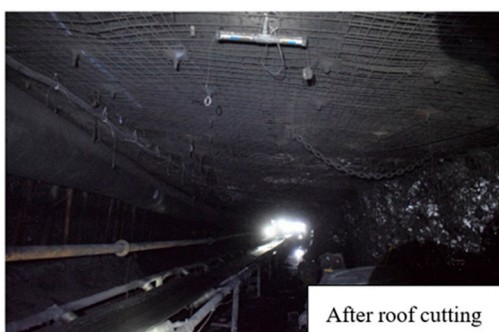

**Figure 12.** Control effect diagram.

## 7. Conclusions

(1) Based on the geological conditions of A110607 return airway, the construction design of roadway excavation with roof-cutting and pressure relief is put forward, and the theory of gob-side entry driving with roof-cutting and pressure relief is analyzed. By cutting off the side hanging roof of goaf, the fracture position of key rock mass is transferred from the top of original roadway to the top of goaf, so as to improve the stress environment of surrounding rock and reduce the deformation of roadway.

(2) Taking the geological conditions of A110607 return airway as the engineering background, through numerical simulation and theoretical analysis, the influence of different roof-cutting height and coal pillar width on the stability of surrounding rock is analyzed from the perspective of roof-cutting effect and surrounding rock deformation.

(3) Through the field blasting test, it is determined that the blasthole spacing is 800 mm, the 321 charging structure is used in complete roof position, and the 221 or 211 charging structure is used in the broken roof and geological structure zone.

(4) The field application shows that the amount of roof and floor movement in the roof-cutting section of the roadway excavation during driving and strong mining is about 38% lower than that in the uncut section, and the deformation of the two sides of the roadway is about 44% lower than that in the uncut section. It shows that the collaborative control scheme of "roof-cutting pressure relief + anchor rope combined support" has a good effect on the roadway excavation driving of small coal pillars in strong mining.

**Author Contributions:** Conceptualization, Z.M.; methodology, M.W.; software, H.Z.; validation, H.M.; formal analysis, M.W.; investigation, X.F.; writing—original draft preparation, M.W.; Formal analysis, M.W. All authors have read and agreed to the published version of the manuscript.

**Funding:** This study was funded by the National Natural Science Foundation of China (No.52274116); Science and Technology Support Plan of Guizhou Province (No. Qian Ke He Zhi Cheng [2021] General 352); Guizhou Science and Technology Plan Project (No. Qiankehe Talent platform [2021]5610); Coal Mine Intelligent Engineering Technology Research Center (No. Qian Ke Zhong Yin Di [2021]4005); Science and Technology Project of Guizhou Province (No. [2021]3001); Guizhou Provincial Department of Education Science and Technology Top Talents Project (No. Qian Jiao Ji [2022] No.072).

**Institutional Review Board Statement:** Not applicable.

**Informed Consent Statement:** Informed consent was obtained from all subjects involved in the study.

**Data Availability Statement:** This study did not report any data.

**Acknowledgments:** All individuals have consented to the acknowledgement.

**Conflicts of Interest:** The authors declare no conflict of interest.

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
