# Peer review of "Control Technology of Roof-Cutting and Pressure Relief for Roadway Excavation with Strong Mining Small Coal Pillar"

_sustainability, doi:10.3390/su15032046_

Round 1

Reviewer 1 Report

This is an interesting paper that covers the subject theoretically and numerically. Results of this study provide some valuable insight into the failure mechanism and control method of underground roadways.  The English language definitely needs to be improved. The important question that needs to be addressed is the following: the key parameters of roof cutting and pressure relief control technology for roadway excavation with strong mining-induced small coal pillars were explored using UDEC code. However, the applied geo-stress on the model is not clear. Besides, how the deformation parameters of the roadway supported by the new proposed support scheme are measured. Please state them in detail.

Reviewer 2 Report

This manuscript discusses the control technology of roof cutting and pressure relief for roadway with small coal pillar. This topic is useful for mining engineering. There are some points in the manuscript that need to be further improved.

1. The section of introduction should be improved significantly to review more associated references to show the necessity of this study.

2. In the present versions of manuscript, the geomechanical parameters of contacts in UDEC model have not been given.

3. The authors should describe how to obtain and determine the geomechanical parameters of rock elements and contacts in the numerical models.

4. What is the basis to determine the fractured lengths of strata over the coal seam?

5. The authors should use the field observation data to confirm the availability of numerical model.

6. Reference format should be modified according to journal requirements.

7. Please add text description at the crack in Fig.9.

8. Part of the image size is too large or too small, need to be adjusted.

9. Some pictures in the article are too large or too small, and need to be adjusted.

10. Chart format needs to be adjusted.

11. Some expressions in this paper are wrong, such as overview of roadway section 1.1 (trapezoidal roadway width × medium height = 5.2 × 2.8).

Reviewer 3 Report

This paper presents

In order to solve the problem of serious deformation and failure of surrounding rock and difficult maintenance of gob-side entry with strong mining-induced small coal pillars, taking the A110607 return airway of Shanwenjiaba Coal Mine as the engineering background, the key parameters of roof cutting and pressure relief control technology for roadway excavation with strong mining-induced small coal pillars were studied by using two-way concentrated blasting roof cutting and pressure relief technology, combined with theoretical analysis, numerical simulation and field industrial test. A collaborative control scheme of ' roof cutting pressure relief + anchor cable combined support ' is proposed. The test results show that when the height of roof cutting is 8m, the angle of roof cutting is 15 °, and the width of coal pillar is 3m, the effect of roof cutting and pressure relief is the best; Through the field blasting test, it is determined that the blast hole spacing is 800mm, 321 charge structure is used in the intact roof, and 221 or 211 charge structure is used in the broken roof and geological structure zone; During the driving and strong mining period, the roof and floor movement of the roof cutting section of the roadway excavation is reduced by about 38 % compared with the uncut section, and the deformation of the two sides of the roadway is reduced by about 44 % compared with the uncut section. It shows that the collaborative control scheme of ' roof cutting pressure relief + anchor cable combined support ' has a good effect on the roadway excavation driving of small coal pillars in strong mining.

1.      Title of paper is looking long, make it concise

2.      Highlight main novelty of your work

3.      Comparison of your current results with the existing literature to prove your novelty and contribution to the research

4.      Applications of this work?

5.      For numerical simulation for part mesh independence? Optimal mesh selected for solution? Mesh statistics

6.      Validation of results? If you are doing your own experiment still show your results matches within 10% error for experiment and simulation parts

7.      Compare your contours/surface plots with experimental images to verify your results are similar in pattern or trend

8.      Add more references to your work

9.      Overall paper is looking short add some more data and results

10.   Conclusion must be revised, and it will be aligned with the abstract and results

Round 2

Reviewer 2 Report

The revised manuscript has been improved well according to the comments.

Reviewer 3 Report

I think the authors addressed all my comments and I recommend it for possible publication in this journal.